# Triple-Antibiotic Combination Exerts Effective Activity against *Mycobacterium avium* subsp. *hominissuis* Biofilm and Airway Infection in an In Vivo Murine Model

**DOI:** 10.3390/antibiotics13060475

**Published:** 2024-05-22

**Authors:** Elliot M. Offman, Amy Leestemaker-Palmer, Reza Fathi, Bailey Keefe, Aida Bibliowicz, Gilead Raday, Luiz E. Bermudez

**Affiliations:** 1Certara, Radnor, PA 19087, USA; offmane@certara.org.ca; 2Department of Biomedical Sciences, College of Veterinary Medicine, Oregon State University, Corvallis, OR 97331, USA; amy.palmer@oregonstate.edu (A.L.-P.); keefeb@oregonstate.edu (B.K.); 3RedHill Biopharma Ltd., Tel Aviv 6473921, Israel; reza.fathi@redhillbio.com (R.F.); radayg@redhillbio.com (G.R.); 4Department of Microbiology, College of Science, Oregon State University, Corvallis, OR 97331, USA

**Keywords:** nontuberculous mycobacteria, *Mycobacterium avium*, biofilm, clarithromycin, mouse model, lung infection

## Abstract

Objectives: Slow-growing nontuberculous mycobacteria (NTMs) are highly prevalent and routinely cause opportunistic intracellular infectious disease in immunocompromised hosts. Methods: The activity of the triple combination of antibiotics, clarithromycin (CLR), rifabutin (RFB), and clofazimine (CFZ), was evaluated and compared with the activity of single antibiotics as well as with double combinations in an in vitro biofilm assay and an in vivo murine model of *Mycobacterium avium* subsp. *hominissuis* (*M. avium*) lung infection. Results: Treatment of 1-week-old biofilms with the triple combination exerted the strongest effect of all (0.12 ± 0.5 × 10^7^ CFU/mL) in reducing bacterial growth as compared to the untreated (5.20 ± 0.5 × 10^7^/mL) or any other combination (≥0.75 ± 0.6 × 10^7^/mL) by 7 days. The treatment of mice intranasally infected with *M. avium* with either CLR and CFZ or the triple combination provided the greatest reduction in CLR-sensitive *M. avium* bacterial counts in both the lung and spleen compared to any single antibiotic or remaining double combination by 4 weeks posttreatment. After 4 weeks of treatment with the triple combination, there were no resistant colonies detected in mice infected with a CLR-resistant strain. No clear relationships between treatment and spleen or lung organ weights were apparent after triple combination treatment. Conclusions: The biofilm assay data and mouse disease model efficacy results support the further investigation of the triple-antibiotic combination.

## 1. Introduction

Nontuberculous mycobacteria (NTMs) are associated with disease in individuals with chronic lung pathology, such as cystic fibrosis, emphysema, and bronchiectasis [1]. The majority of the NTM infections are linked to environmental exposure [2]. Many of the NTMs, however, are in fact opportunistic pathogens in animals and humans causing severe disease. Person-to-person transmission is relatively uncommon, and human disease caused by *M. avium* subsp. *hominissuis* more frequently occurs via environmental (water, soil) exposure in susceptible immunocompromised hosts. Nonetheless, the specific source of infection often goes undetected [2,3].

Pulmonary manifestations account for 80–90% of all NTM-associated disease in non-AIDS patients [4], and approximately 80% of pulmonary NTM infections in the U.S. are caused by the *M. avium* complex [5,6]. Patients with pulmonary NTM disease present with nonspecific symptoms, often indistinguishable from tuberculosis or other respiratory diseases. Typically, patients have a chronic or recurring cough but also present other respiratory symptoms such as sputum production, hemoptysis, and/or dyspnea [4]. Patients commonly present with fatigue, fever, weight loss, asthenia, and/or anorexia [4,7].

NTM infection of the lung is commonly associated with the presence of biofilms in the airway, making the clinical and microbiological response to the therapy a challenge [8,9]. Antibiotics with strong in vitro activity might fail to eliminate biofilms as source of infection [10]. Without successful treatment, *M. avium* lung disease is characterized by progressive, irreversible lung damage and increased mortality [11]. Risk factors related to *M. avium* lung disease progression include a low body mass index (BMI), poor nutritional status, presence of cavitary lesions, extensive lung disease, and/or positive acid-fast bacilli in the sputum. The emergence of macrolide resistance on current treatment options has the potential to result in devastating outcomes. In fact, a study detected a 34% mortality rate in patients with macrolide-resistant *M. avium* lung disease who remained culture positive after 1 year [4].

The therapy of *M. avium* disease is based on the efficacy of macrolides (clarithromycin and azithromycin), and the synergistic effect of ethambutol, demonstrated both in experimental models [12] and in clinical trials [13]. The use of additional antimicrobials such as rifampin, are recommended, largely to mitigate the emergence of macrolide resistance, as opposed to being used to reduce the microbial burden. Other compounds such as clofazimine, have been suggested to have activity against *M. avium* and are currently being evaluated for clinical benefit in NTM infections [14,15].

The prolonged time for treatment increases the chances of a lack of compliance. The ingestion of many different pills several times a day also has a deleterious effect, with a great possibility of developing stomach mucosa intolerance. Many years ago, the world health organization developed a combined therapy for tuberculosis to establish the medication as one pill a day, containing the main compounds active against *Mycobacterium tuberculosis*, isoniazid, rifampin, and pyrazinamide. Physical tolerance is considered one of the problems with the need for ingestion of many antibiotics, but an additional consequence is the potential for forgetting and missing the schedule for the administration of the medication, which can be associated with an increase in resistant bacteria.

To address the aspect that the therapy of *M. avium* infection can take 18 months, RHB-204 (ClinicalTrials.gov identifier: NCT04616924) is being developed as fixed-dose oral capsule containing a combination of clarithromycin (CLR), rifabutin (RFB), and clofazimine (CFZ), for treating NTM lung disease caused by *M. avium* lung infection. RHB-204 is designed to mitigate the emergence of macrolide resistance, observed with macrolide monotherapy. Clarithromycin is an effective anti-*M. avium* complex medication that has been used clinically for many years. Clofazimine is a compound used in the treatment of leprosy that has been evaluated for the treatment of *M. avium* infection, with ongoing clinical trials. In experimental animal models, it has been shown to have a synergistic effect with clarithromycin for the therapy of *M. avium* infection [15]. This report describes the antibacterial activity of the triple combination in an in vitro biofilm assay and an in vivo murine model of *M. avium* lung infection. Clarithromycin, clofazimine, and rifampin mixed in a compound showed efficacy against *M. avium* in a mice system. The therapy concentrated on the administration of one pill a day, which may provide superior adherence from patients, and it has the potential to be associated with a decrease in the emergence of bacterial resistance during treatment.

## 2. Results 

### 2.1. The Combination of CLR, RFB, and CFZ Is More Effective Than a Single Antibiotic or a Double Combination against Bacterial Biofilms

We assessed the activity of CLR-RFB-CFZ in an in vitro *M. avium* biofilm assay [16,17,18]. The biomass and colony forming units (CFUs) of the microorganisms after various treatments relative to the initial inoculum of strains MAC 104 and MAC A5 were determined. By four and seven days of clinically relevant drug exposure, the CFU/mL was significantly lower in the triple combination as compared to the single drug or two-drug combinations, thus indicating that the triple combination treatment is superior in terms of anti-*M. avium* activity against organisms in biofilm matrices (Table 1). Table 1 shows the results obtained with strain MAC A5. The efficacy against MAC 104 biofilms was similar.

### 2.2. CLR-RFB-CFZ Is Effective in an In Vivo Murine Model of M. avium Infection with CLR-Susceptible and CLR-Resistant Strains of M. avium

To determine whether the triple combination of CLR-RFB-CFZ is increasingly active as compared to single antibiotics or other combinations against the bacterium in vivo, we examined whether CLR-RFB-CFZ has activity against *M. avium* in a murine infection model. Six-week-old C57BL/6 mice were infected intranasally with 2 × 10^7^ CFU/0.1 mL of either MAH 104 CLR-sensitive or CLR-resistant MAC 101-R, *M. avium*. Following four- and eight-week treatment periods, the mice were sacrificed, and their lungs and spleens were harvested for CFU/organ determination. The CFU reductions after four and eight weeks of treatment relative to baseline and vehicle-treated animals are presented after infection with CLR-sensitive or CLR-resistant strains, respectively. In the CLR-sensitive experiments, none of the bacterial colonies were found to be resistant to the triple combination or double combination with CFZ and CLR (Table 2).

In both the four- and eight-week experiments, the baseline group animals ± vehicle exhibited infection with a high bacterial burden in both the lungs and the spleen. In MAH 104-infected mice, relative to the vehicle, CLR alone resulted in an approximately 1- and 2-log unit reductions in CFU after four weeks of treatment in the lung and spleen, respectively (Figure 1 and Figure 2). The mean response of the triple combination was comparable to that of the double CLR +CFZ combination and performed superiorly in terms of reducing lung tissue CFUs when compared to each of the single-drug treatments (i.e., CLR, RFB, or CFZ alone) or double combinations with RFB. Further, the distribution of individual animal data supports a stronger response from the triple combination than from any double combination as evidenced by some animals in these treatment groups exhibiting multiple log reductions or no evidence of lung infection at all as measured by lung CFUs/organ. Results in the splenic tissue followed similar trends to those in the lung with respect to treatment effects, although the baseline CFU/organ counts were on average 3 log units lower in the spleen three weeks following lung inoculation.

### 2.3. Emergence of CLR-Resistant Strains in Mice

In the CLR-sensitive experiments, one animal in the CLR monotherapy group had 200 CFUs resistant to CLR. When this bacterium was plated out with agar containing a combination of RFB and CLR, the number of resistant CFUs/organ was reduced by half to 100 CFUs. Additionally, none of the CFUs/organ were found to be resistant to the triple combination or double combination with CFZ and CLR. Out of twenty mice receiving the combination of CLR and RFB or CLR and CFZ, none had any resistant CFUs/organ.

The natural frequency of resistance of MAC 104 to CLR was determined to be 1 × 10^−9^. The surviving CFUs/organ from the lung homogenates of both MAC 104 and MAC 101-R were plated on antibiotic agar containing the MICs of the drugs (Table 2). Four and eight weeks of treatment with CLR monotherapy resulted in a 0.129% and 1.129% increase, respectively, compared to the control. Eight weeks of treatment, however, with additional RFB or CFZ induced a reduction of the proportion of the resistant colonies by more than 50%, to less than 0.47%, while the triple combination resulted in no resistant colonies being detected.

### 2.4. Lung Histopathology of Mice Infected with MAC ± after Four and Eight Weeks of Treatments (±CLR, ±RFB, ±CFZ Combinations) Reveals the Greatest Improvement for Treatment with the Triple Combination CLR-RFB-CFZ

Biopsies with histopathologic features consistent with mycobacterial infections (e.g., granulomatous inflammation or positive AFB stain) are routinely used to diagnose and evaluate clinical *M. avium* complex infection [4]. To evaluate the effectiveness of CLR-RFB-CFZ, we performed histopathological analysis by cutting eight slices per lung in pairs of mice. The slices were evaluated in a blinded fashion. Pre-inoculation, the lung histology was normal without any pathological features. The lung and spleen histopathology findings for each of the treatment groups are shown in (Table 3). Treatments with the double and triple combinations were associated with significant histopathologic improvement in the lung.

### 2.5. Pharmacokinetic Parameters Confirm Drug Exposure Is Consistent with the RHB-204 Clinical Regimen Being Tested in Clinical Trials

The pharmacokinetic parameters (of area under the curve over the dosing interval [AUC_TAU_], maximum concentration [C_max_], and steady-state average concentration [C_avg_]) were derived for the composite concentration versus time profiles (n = 2 mice per timepoint) in each treatment group by noncompartmental analysis (Phoenix ^®^ WinNonlin^®^, Certara, Clayton, MO, USA).

CLR exposure in mice resulted in comparable PK parameters after 4- and 8-weeks of treatment. When combined with RFB or CFZ, CLR was on average lower after 8 weeks of treatment when compared to 4 weeks of treatment. These results suggest that, in mice, the inductive effects of RFB and/or CFZ on CLR metabolism may not have fully achieved a steady state after four weeks of treatment. Regardless, across treatments, CLR administered once daily to mice with *M. avium* resulted in an average concentration over the dosing interval ranging from 78 ng/mL to 438 ng/mL. No marked difference was observed in CFZ PK after 4 or 8 weeks of treatment when administered alone. Repeated daily administration of CFZ concomitantly with CLR or CLR and RFB for 4 or 8 weeks resulted in a markedly higher exposure when compared to CFZ administered alone, likely reflecting the impact of CLR’s inhibitory effect of CFZ metabolism and/or transport. Nonetheless, the C_avg_ ranged from 211 to 500 ng/mL.

Animals treated with RFB alone for eight weeks displayed lower systemic exposure to RFB compared to those animals treated for four weeks. The precise mechanism accounting for this is not clearly understood; however, this may be in part due to ongoing auto-induction as RFB is an inducer and substrate for CYP3A4. The C_avg_ values ranged from 128 ng/mL to 330 ng/mL across all the treatments tested.

## 3. Materials and Methods

### 3.1. Bacterial Strains

*Mycobacterium avium* subsp. *hominissuis* (*M. avium*) (MAC strains 104, A5, and 101-R [clarithromycin resistant strain in the in vivo mouse experiments] were isolated from the blood and lung of patients. MAC 104 was isolated from a patient at UCLA, while MAC A5 was obtained from a patient in England and provided by Dr. Katherine Eisenach (University of Arkansas), and MAC 101-R was obtained from a patient at UCLA. Clarithromycin resistance was confirmed via sequencing and determined to have a single-base mutation in the 23S RNA gene (A2275 to a C2275) that confers resistance to CLR (MIC > 128 μg/mL). Bacterial suspensions were generated in Hanks’ balanced salt solution (HBSS, Corning, Teksbury, MA, USA), and the optical density was used to obtain 3 × 10^8^ CFU/mL (confirmed by OD). Inoculum stocks of *M. avium* were resuspended in Hanks’ buffered salt solution (HBSS) and passed through a 23-gauge needle ten times to disperse clumps; the dispersion of the inoculum was confirmed microscopically and quantified against the 1.0 McFarland turbidity standard prior to dilution. The inoculums were diluted and plated onto Middlebrook 7H10 agar (VWR, Radnor, PA, USA) to confirm the concentration of bacteria prior to and use for in vitro and in vivo experiments.

### 3.2. Antimicrobial Agent Preparation

Test materials received from RedHill Biopharma Ltd. (Tel Aviv, Israel) were stored at 4 °C until prepared in sterile saline containing 10% DMSO and sonicated for 10 min for oral gavage. These included 99.2% purity RFB, 99.5% purity CFZ, and 99.7% purity CLR (powder preparations) stored after preparation at 21 °C and protected from light. The preparations were made every day before administration to mice.

### 3.3. Antimicrobial Susceptibility Testing

Susceptibility testing was conducted using single- and two-fold broth microdilutions as described in the literature [19,20,21,22]. The suspensions were prepared by swabs on a growth plate with a sterile cotton swab or were prepared in broth culture. The broth was transferred to 4.5 mL of phosphate buffer solution (PBS) until the turbidity matched the density of a 0.5 McFarland standard as confirmed by OD. The suspensions were then vigorously mixed in a vortex mixer for 15–20 s. The final inoculum was prepared to a bacterial density of 10^5^/mL. The tube was inverted 8–10 times to mix the suspension. In mice experiments, the antibiotics were used in the following dosages: clarithromycin, 50 mg/kg; rifabutin, 10 mg/kg; clofazimine, 10 mg/kg.

### 3.4. Biofilm Methodology

In brief, biofilm experiments were conducted for both MAC 104 and MAC A5 strains. The MAC A5 strain was used due to the fact that its biofilm has significantly more extracellular DNA (eDNA) than the MAC 104 biofilms [8]. The 7H10 agar plate(s) were streaked with low-passage MAC 104 and A5, allowing for one week of log phase growth. Bacterial suspensions in HBSS were diluted in 16 mL of HBSS to make a suspension of 1 × 10^8^ bacteria/mL. The tubes were then diluted in HBSS 1:10 and had an absorbance optical density (O.D.) of 0.018–0.021 (10^6^/mL). The contents of the bacterial suspension were then poured into a reservoir and allowed to settle for 5 min, at the bottom of the reservoir. Using a multichannel pipette, 150 μL of bacterial suspension was then transferred into a flat-bottomed 96-well plate (VWR). The plates were placed into a sealable plastic bag (Ziploc, San Diego, CA, USA) to prevent evaporation, and the biofilms were allowed to grow in the dark for one week at 25 °C.

To assess the impact of each drug component, the medium was replaced with 150 μL of fresh HBSS containing the desired concentrations of antibiotics and transferred to 37 °C. The concentrations for the biofilm assessments were determined first by estimating the average concentration at steady state for each drug component when administered in combination for the treatment of *M. avium* lung infection and multiplying by an estimated epithelial lining fluid (ELF) concentration to total plasma concentration ratio as stated above. The penetration of CLR into the epithelial lining fluid (ELF), i.e., ELF/plasma ratio, is relatively well documented and has been reported to be approximately 3.5-fold to as high as 40-fold at clinically relevant doses ranging from oral 200 mg single doses to 500 mg twice daily [23,24]. As ELF is largely free of plasma proteins, the free average ELF concentration at the airway–biofilm interface is 3.4–7.8-fold greater than the average total concentration at steady state in plasma [25]. For RFB, there are little data to inform the ELF/plasma ratio; however, ratios of approximately 0.34 have been reported for rifampin [26]. For CFZ, there are similarly little experimentally derived data to support the ELF/plasma ratio; however, Dheda (2018) obtained airway and blood concentration data in a limited group of individuals, which resulted in an estimated ELF/plasma ratio of approximately 0.2% [26].

The antibiotic-treated plates were placed back in the plastic bag and incubated at 37 °C in the dark for seven days. Following the incubation period, the contents of the wells in the plate were disrupted with vigorous pipetting. The suspension was then serially diluted in HBSS from 10^−3^ to 10^−5^ and plated onto 7H10 agar plates. Bacteria were allowed to grow for 10 days before determining the surviving colony forming units (CFU)/mL. Pairwise comparisons were made and determined by the Mann–Whitney nonparametric test (GraphPad Prism version 8.0.1, Boston, MA, USA).

### 3.5. Animal Experiments

All the performed experiments were carried out according to the guidelines for animal ethics. All the experiments were reviewed and approved by the IACUC committee of Oregon State University (ACUP#4257). Six-week-old C57BL/6 mice (approximately 20 g) were purchased from Jackson Laboratory (San Diego, CA, USA) and delivered to the Laboratory Animal Research Center at Oregon State University. The mice were housed in sterilized, individually ventilated cages always exposing the mice to HEPA-filter sterile air. The mice had free access to food and water (sterile) and were provided with aspen chip bedding. The room temperature was 22 ± 1 °C, with a relative humidity of 30%. The mice were exposed to 12 h light/dark cycles.

Bacteria (MAC 104) were prepared for infection by resuspension in HBSS to concentrations of 3 × 10^8^ CFU/mL. Prior to the intranasal infection of mice, the suspension was vortex agitated for two minutes and passed through a 23-gauge needle ten times to disperse clumps. Microscopic observations confirmed the dispersion of the initial inoculum. The suspensions were serially diluted and plated onto 7H10 agar to confirm the CFU/mL of the inoculum. The mice were infected intranasally with 2 × 10^7^ CFU/mL of MAC 104 or CLR-resistant MAC 101-R. To minimize sneezing or aspiration, each mouse was lightly sedated with brief exposure to isoflurane (Vet One, 3% isoflurane/97% air) prior to inoculation. The infection was allowed to establish for three weeks. The mice were treated orally with test antibiotics once a day (QD) for up to four or eight weeks after the establishment of the infection. This report details the results of the CLR-sensitive and CLR-resistant, four-week and eight-week treatments. For each strain, the mice were allocated (volume 0.1 mL) to one of eight groups and treated for four weeks: vehicle, CLR, RFB, CFZ, CLR + RFB, CLR + CFZ, CLR + CFZ + RFB, or baseline. The antibiotics were administered at the following doses: clarithromycin (CLR), 50 mg/kg; rifabutin (RFB),10 mg/kg; clofazimine (CFZ), 10 mg/kg.

In each of the four-week treatment groups, the treatments were administered once daily. The weight and the clinical conditions of the mice were recorded on Tuesdays and Fridays. The mice were euthanized using the CO_2_ method. Their lungs and spleens were aseptically excised and homogenized using a Bead Ruptor Elite (OMNI international, Kennesaw, GA, USA) following the manufacturer’s protocol for bacterial isolation from host tissue. Homogenized samples were then serially diluted and plated onto 7H10 agar plates for CFU determination. The total viable bacterial count was determined following 10 days of incubation at 37 °C.

The total bacterial viable counts (CFU/mL of organ homogenate) were expressed per lung or spleen tissue (CFU/organ). Activity against the *M. avium* strains was tabulated and summarized descriptively by treatment group and strain. The descriptive statistics included the sample size (N), mean, and standard deviation (SD). Bar plots were produced to illustrate the side-by-side treatment effects with SD error bars and stratified by strain. Individual tables and figures were produced for the lung and spleen. The treatment groups were compared using a one-way ANOVA to determine variance between the groups. Then, the treatment groups were independently compared to both baseline and vehicle via a one-way ANOVA at an alpha of 0.05. The tables and plots were produced using R and GraphPad Prism (version 8.0.1).

The lung tissues were fixed and stained with H&E and acid-fast staining as previously reported [19,20,21,27].

To determine the antibiotic levels after treatment, the mice receiving antibiotics were bled at several timepoints, and the antibiotic concentrations were measured (two mice per timepoint) by noncompartmental analysis.

### 3.6. Emergence of Resistance Experiments

Lung tissue was removed from each treated animal and cultured to determine the number of viable bacteria present. The tissue samples were plated with and without antibiotic on 7H10 agar. Antibiotic plates were made with minimum inhibitory concentrations (MICs) of 8 μg/mL for CLR and 0.25 μg/mL for both RFB and CFZ. The plates were made for each of CLR alone, CLR + RFB or CLR + CFZ, and the triple combination CLR + RFB + CFZ. All the plates were incubated in a microbiological incubator at 37 °C for 14 days.

A second set of experiments was conducted to assess the contribution of RFB and CFZ in mitigating the selection of resistance when administered as a dual-agent treatment with CLR or as a triple combination treatment. In this experiment, inoculums from the harvested lungs were plated on Middlebrook 7H10 agar plates at the MICs of each drug. The homogenates were diluted from 10^−2^ to 10^−6^ and plated. The observed colonies were tested for MIC using the methods described above and in References [1,2,3]. In addition, a range of inoculums of MAC 104 (10^11^ to 10^5^/mL) were incubated with the MIC of the antibiotics in 7H9 broth for seven days and plated onto 7H10 agar plates to determine the frequency of resistance before host infection.

### 3.7. Statistical Analysis

The total bacterial viable counts (CFU/mL of homogenate) were expressed per lung or spleen tissue (CFU/organ). Activity against the MAC strains was tabulated and summarized descriptively by treatment group and strain. The descriptive statistics included the sample size (N), mean, and standard deviation (SD). Bar plots were produced to illustrate the side-by-side treatment effects with SD error bars and stratified by strain. Individual tables and figures were produced for the lung and spleen. The treatment groups were compared using a one-way ANOVA to determine variance between the groups. Then, the treatment groups were independently compared to both baseline and vehicle via a one-way ANOVA at an alpha of 0.05. The tables and plots were produced using R and GraphPad Prism (version 8.0.1, Boston, MA, USA), and statistical analysis was performed using GraphPad Prism.

## 4. Discussion

Currently, therapy for *M. avium* lung infection is an exceptional clinical challenge because of the lack of persistent efficacy and treatment safety concerns. The ability to resist intracellular killing and to form robust biofilms can impart antibiotic resistance to NTMs like *M. avium* [28]. NTM infections can lead to a progressive decrease in lung function. The investigational drug formulation RHB-204, which is an all-in-one fixed-dose combination of CLR-RFB-CFZ, has the potential to address persistently emergent antibiotic resistance among *M. avium* strains. These active ingredients and doses were selected to maximize therapeutic effectiveness against MAC, while, perhaps, minimizing the risk of toxicity, particularly as related to the RFB and CFZ components. 

In this study, we evaluated the anti-*M. avium* activity of CLR, CFZ, and RFB as individual drug treatments and in double- and triple-drug combinations. The triple-drug components of RHB-204 were assessed with respect to action against bacteria in biofilms, *M. avium* load after lung/spleen infection, and emergence of CLR-resistant strains. Recent studies have demonstrated that *M. avium* on exposure to stress as well as antibiotics responded with significant changes in protein synthesis, inducing adaptations and survival of the microbe under the new environmental conditions [29].

The triple-drug combination treatment CLR-RFB-CFZ against in vitro *M. avium* biofilms was superior to any individual component or double combination. Since this is one of the bacterial phenotypes encountered in the lungs of patients, where an extracellular biofilm matrix in the airways is formed, we conclude that this approach has the potential to provide sustained benefit for patients with established *M. avium* biofilms. In addition, the treatment of mice infected with CLR-susceptible *M. avium* resulted in significant reductions in bacterial counts in the lung after four- and eight-weeks of treatment. None of the CFUs in the CLR-susceptible strains were resistant to the three-drug combination or two-drug combination with CFZ and CLR. Moreover, out of twenty mice receiving the combination of CLR and RFB or CLR and CFZ, none had any resistant CFUs. Most significantly, the combination of RFB and CFZ together with a CLR backbone regimen mitigated the emergence of CLR-resistant *M. avium* strains and responses to therapy. 

The doses selected in the current study were intended to reflect the clinically relevant doses of each drug in the RHB-204 investigational drug product. The dose response of CLR in *M. avium* has been previously evaluated in beige mice, at doses ranging from 50 to 300 mg/kg/day orally (intravenous [IV] inoculation of *M. avium*). CFU counts in the spleen, liver, and lung suggest a flattening of the response between the 200 mg/kg and 300 mg/kg dose in the spleen and liver CFUs, suggesting that 200 mg/kg/day orally is associated with the maximal response in disseminated models of mouse infections. Bermudez [12] confirmed the efficacy of CLR 200 mg/kg in disseminated MAC infections in mice when comparing CLR and azithromycin (200 mg/kg/day). Alone, both macrolides were effective at reducing spleen and lung CFUs after intravenous (IV) *M. avium* inoculation, with CLR being slightly more effective but with resistance to CLR emerging more frequently. 

CLR has also been demonstrated to be effective in *M. avium* infections when the inoculum was administered via the intranasal route causing lung infections [27]. CLR 100 mg/kg per day or administered twice weekly, reduced lung CFUs, although to a lesser extent for biweekly than for daily, when compared to the control group. While the PK of CLR was not reported in these studies, the systemic exposure of CLR at doses above 50 mg/kg is anticipated to exceed that associated with the recommended doses (i.e., 500 mg BID) of CLR in humans treated for *M. avium*. RHB-204 is currently being studied for the treatment of pulmonary MAC disease in an ongoing Phase 3 clinical trial (NCT04616924). In this ongoing trial, the treatment regimen provides for 475 mg of CLR, twice daily, three times a week. The CLR C_avg_ in humans is anticipated to be within the range of the reported C_avg_ for CLR in mice treated with 50 mg/kg (unpublished data).

Similarly, the C_avg_ values observed in mice treated with CFZ 10 mg/kg and RFB 10 mg/kg are anticipated to be within the range of C_avg_ in humans treated with RHB-204, treated with 40 mg and 120 mg of CFZ and RFB, respectively, twice daily, three times a week. Thus, the doses selected in the current murine model of NTM disease reflect therapeutically relevant systemic exposure in humans.

Examinations of the mouse body weight as well as the lung and spleen organ weights were also performed over the course of this study (unpublished observations). For all the experiments, the mice continued to gain weight, on average, with no discernable trend between treatments or durations of experiment. Increases in the lung or spleen organ weights could potentially occur due to inflammatory responses and increased bacterial burden. With effective antimicrobial treatment, the inflammatory processes are anticipated to be reduced, and the tendency is to observe decreased organ weights with ongoing effective treatments compared to those for treatments that are less effective. However, a direct correlation may not universally be observed due to fibrotic changes in the organs that may not be reversible with treatment. Thus, we examined the lung and spleen weights ± 4 or 8 weeks of treatment to evaluate whether organ weights suggestive of inflammation were altered in any way by single-, double-, or triple-drug antibiotic treatments. No clear relationship between treatment and body or organ weights was apparent for either spleen or lung weights.

Collectively, the biofilm and animal data demonstrated a significant response to the three-drug combination as compared to any single drug or two-drug combination. Clofazimine has been shown to be active against *M. avium* in lung models, either alone or in combination with CLR [15,30]. The three-drug combination in which clofazimine and rifabutin are added to a clarithromycin backbone appears to provide the greatest protection against the emergence of macrolide resistance and thus represents an important treatment option in the armamentarium against *M. avium* infection. 

The adoption of this strategy for the treatment of patients with *M. avium* lung infection may have long-term benefits, with easier adherence to the therapy schedule, decrease in the emergence of resistance to macrolides caused by association with rifabutin, and consequently less chance for *M. avium* to develop resistance to the therapy by the erratic intake of medication.

## Figures and Tables

**Figure 1 antibiotics-13-00475-f001:**
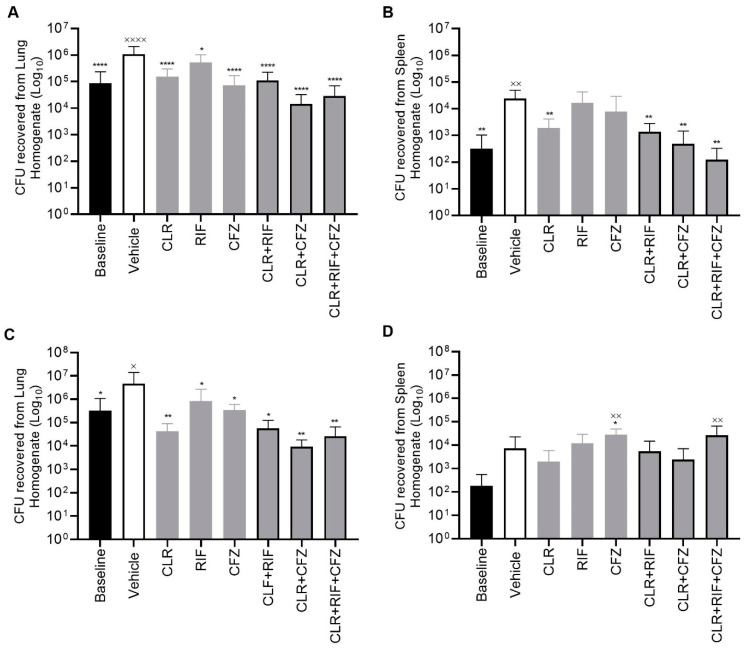
Organ homogenates from chronic lung infection with a CLR-sensitive strain in a C57BL6/J mouse model after treatment with clofazimine, 10 mg/kg (CFZ); clarithromycin, 50 mg/kg (CLR); rifabutin, 10 mg/kg (RIF); or in combination. After treatment with the indicated compounds, the mice were euthanized and the organs harvested. (**A**) CFU levels of lung homogenates after 4 weeks of treatment and (**B**) CFU levels of spleen homogenates after 4 weeks of treatment. (**C**) CFU levels of lung homogenates after 8 weeks of treatment and (**D**) CFU levels of spleen homogenates after 8 weeks of treatment. All treatment groups consisted of 10 mice. One-way ANOVA test for statistical analysis of treatment groups compared to baseline ^x^ *p* < 0.05, ^xx^ *p* < 0.001, ^xxxx^ *p* < 0.0001 and vehicle * *p* < 0.05, ** *p* < 0.001, **** *p* < 0.00001. CFU = colony forming unit.

**Figure 2 antibiotics-13-00475-f002:**
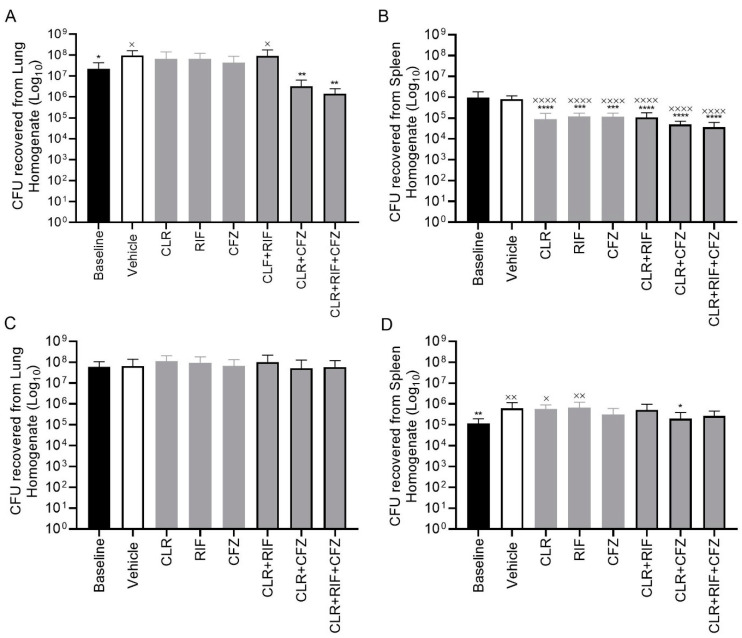
Organ homogenates from chronic lung infection with the CLR-resistant strain in a C57BL6/J mouse model after treatment with clofazimine, 10 mg/kg (CFZ); clarithromycin, 50 mg/kg (CLR); rifabutin, 10 mg/kg (RIF); or in combination. After treatment with the indicated compounds, the mice were euthanized and the organs harvested. (**A**) CFU levels of lung homogenates after 4 weeks of treatment and (**B**) CFU levels of spleen homogenates after 4 weeks of treatment. (**C**) CFU levels of lung homogenates after 8 weeks of treatment and (**D**) CFU levels of spleen homogenates after 8 weeks of treatment. All treatment groups consisted of 10 mice. One-way ANOVA test for statistical analysis of treatment groups compared to baseline ^x^ *p* < 0.05, ^xx^ *p* < 0.001, ^xxxx^ *p* < 0.0001 and vehicle * *p* < 0.05, ** *p* < 0.001, *** *p* < 0.001, **** *p* < 0.00001. CFU = colony forming unit.

**Table 1 antibiotics-13-00475-t001:** Antimicrobial activity against MAC A5 biofilms: individual and with combinations.

	CFU/mL (×10^7^)	
Treatment	4 Days	7 Days
HBSS (negative control)	1.69 ± 0.4 ^a^	5.20 ± 0.5
Clarithromycin ^1^	1.35 ± 0.3	1.07 ± 0.3
Rifabutin ^1^	1.41 ± 0.5	1.64 ± 0.3
Clofazimine ^1^	1.91 ± 0.6	1.42 ± 0.5
Clarithromycin/Rifabutin ^1,2,3^	1.38 ± 0.3	0.75 ± 0.6
Clarithromycin/Clofazimine ^1^	1.33 ± 0.4	1.08 ± 0.5
Clarithromycin/Rifabutin/Clofazimine ^1,2,3,4,5,6^	0.30 ± 0.4	0.12 ± 0.5

CFU = colony forming unit; HBSS = Hanks’ balanced salt solution. ^a^ Mean ± Standard deviation. ^1^ *p* < 0.05 compared to HBSS control. ^2^ *p* < 0.05 compared with clarithromycin treatment. ^3^ *p* < 0.05 compared with rifabutin treatment. ^4^ *p* < 0.05 compared with clofazimine treatment. ^5^ *p* < 0.05 compared with clarithromycin + rifabutin. ^6^ *p* < 0.05 compared with clarithromycin + clofazimine.

**Table 2 antibiotics-13-00475-t002:** Emergence of CLR resistance after four and eight weeks of treatment in mice infected with MAH.

	Total CFU		
	(% of Total CFU Resistant to Clarithromycin) ^4^		
Treatment	In Vivo Resistance	4 Weeks Treatment	8 Weeks Treatment
Clarithromycin	1 × 10^−9^ (0.0001%)	1.55 × 10^5^ (0.129%)	5.31 × 10^4^ (1.129%)
Clarithromycin/Rifabutin		1.11 × 10^5^ (0.09%)	5.7 × 10^4^ (0.473%) ^1^
Clarithromycin/Clofazimine		1.43 × 10^4^ (0.139%)	9.16 × 10^3^ (0.469%) ^1^
Clarithromycin/Rifabutin/Clofazimine		2.85 × 10^4^ (0.035%)	2.60 × 10^4^ (0.0%) ^1,2,3^

^1^ *p* < 0.05 compared with clarithromycin. ^2^ *p* < 0.05 compared with clarithromycin/rifabutin. ^3^ *p* < 0.05 compared with clarithromycin/clofazimine. ^4^ The percentages were obtained by dividing the number of resistant colonies by the total number of colonies × 100.

**Table 3 antibiotics-13-00475-t003:** Histopathology findings after either 4 or 8 weeks of antibiotic treatment.

Treatment Group	
Saline	- Marked multifocal lymphocytic and plasma cell infiltration, diffuse congestion.
	- Dense infiltration of lymphocytes, plasma cells, and macrophages; infiltration of the blood vessels; formation of nodes or granulomas with many bacteria.
	- Bacteria in the tissue, nodes, and mucosal area.
Clarithromycin	- Moderate multifocal lymphocytic and plasma cells in the adventitia of multifocal pulmonary blood vessels, forming nodular aggregates or granulomas with bacteria.
	- Perivascular inflammation, with minimal peribronchiolar lymphocytic infiltration.
	- Diffuse congestion, bacteria in granulomas.
Rifabutin	- Diffuse congestion.
	- Moderate and sometimes marked infiltration of lymphocytes and plasma cells in the adventitia of multiple pulmonary blood vessels, formation of granulomas and aggregates with bacteria.
	- Infiltrates of lymphocytes are also observed in multiple locations in the bronchi and bronchioles.
Clofazimine	- Mild diffuse congestion.
	- Minimal multifocal congestion of lymphocytes and plasma cells.
	- Granulomas in multiple locations with a mild number of bacteria.
	- Infiltration into the bronchiolar walls.
Clarithromycin	- Multidiffuse congestion.
+Clofazimine	- Minimal multifocal infiltration of lymphocytes and plasma cells in the lung tissue and in the bronchiole walls.
	- Limited number of granulomas with few bacteria.
Clarithromycin	- Mild to moderate and sometimes marked infiltration of the adventitia of multiple pulmonary blood vessels with lymphocytes and plasma cells, formation of nodular aggregates with bacteria.
+Rifabutin	- Infiltration in the walls of the airways.
Clarithromycin	- Minimal diffuse congestion.
+Rifabutin	- Minimal multifocal infiltration of lymphocytes, with occasional plasma cells in the bronchial walls.
+Clofazimine	- Rare, resolving granulomas.

## Data Availability

Data are contained within the article.

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
