# Peer review of "Triple-Antibiotic Combination Exerts Effective Activity against Mycobacterium avium subsp. hominissuis Biofilm and Airway Infection in an In Vivo Murine Model"

_antibiotics, 2024, doi:10.3390/antibiotics13060475_

Round 1

Reviewer 1 Report

Comments and Suggestions for Authors

The manuscript describes an antimicrobial activity of a combination of three antibiotics (clarithromycin, rifabutin, and  clofazimine).   The treatment, if performance characteristics are confirmed, might have practical application in treatment of lung infections caused by Mycobacterium avium subsp. hominissuis.

While the study has produced encouraging results, the overall presentation  can be improved to meet academic publication criteria.

Major concerns:

1) The manuscript needs to be  thoroughly reviewed by authors for stylistic consistency (L206-215, Section 2.4, etc.), use of abbreviations (L89, what is MAH?),  repeated statements  ( L175-184 and 206-215, etc.), relatedness of certain statements to the text  (for example, L162-165)

2) The antimicrobial effect on the  MAC biofilm has been evaluated at 25oC,  which is the suboptimal temperature for Mycobacterium avium subsp. hominissuis. Why authors has chosen the  temperature 25oC, while the post-treatment viable bacterial count in animal organs has been done at 37oC? A solid justification is required for this experimental approach.

3)  The antibiotic concentrations, that have been used in the study,  must be clearly stated in the Materials and Methods. The purity of the material provided in the text (99.2 – 99.7%, L98 ) is confusing. Is it the purity of a starting material? The following passage on the concentrations for biofilm assessments (L120-134) does not provide the clear answer on the antibiotic concentrations used in the study, has to be moved to the Discussion section, and overall is confusing,  as the reflections on ELF, plasma , and lung infections contradict the biofilm experiments conducted at 25oC.

4) What author mean by the “vehicle” (L164, 182, 213, etc.) when describing the data analysis, for example “…treatment groups were 181 independently compared to both baseline and vehicle…”? I am not familiar with this terminology.

Comments on the Quality of English Language

The manuscript must be  thoroughly reviewed by all authors for stylistic consistency.

Author Response

Major concerns:

1) The manuscript needs to be  thoroughly reviewed by authors for stylistic consistency (L206-215, Section 2.4, etc.), use of abbreviations (L89, what is MAH?),  repeated statements  ( L175-184 and 206-215, etc.), relatedness of certain statements to the text  (for example, L162-165)

Answers: Thank you for pointing out the problems. The manuscript has been revised for style and consistency as suggested. The abbreviations, with exception to bacterial strains and antibiotics, have been eliminated from the text.

2) The antimicrobial effect on the  MAC biofilm has been evaluated at 25oC,  which is the suboptimal temperature for Mycobacterium avium subsp. hominissuis. Why authors has chosen the  temperature 25oC, while the post-treatment viable bacterial count in animal organs has been done at 37oC? A solid justification is required for this experimental approach.

Answer: The reviewer was correct. There was a confusing statement. It was corrected and state that biofilms were created at 25 C for days specified, but were transferred to 37 C before exposure to antibiotics. The information has been corrected in. the text.

3)  The antibiotic concentrations, that have been used in the study,  must be clearly stated in the Materials and Methods. The purity of the material provided in the text (99.2 – 99.7%, L98 ) is confusing. Is it the purity of a starting material? The following passage on the concentrations for biofilm assessments (L120-134) does not provide the clear answer on the antibiotic concentrations used in the study, has to be moved to the Discussion section, and overall is confusing,  as the reflections on ELF, plasma , and lung infections contradict the biofilm experiments conducted at 25oC.

Answers: a. The concentrations of antibiotics are now stated in the methods section. B. The purity of the materials (antibiotics) were more than 99% as stated, before the dilutions necessary to establish the doses. C. The paragraph related to the biofilm we believe that the point should be made in the methods used.

4) What author mean by the “vehicle” (L164, 182, 213, etc.) when describing the data analysis, for example “…treatment groups were 181 independently compared to both baseline and vehicle…”? I am not familiar with this terminology.

Answer: The vehicle terminology  refers to the diluent in which the antibiotic powder was suspended. As far we understand, it is a common used terminology.

Thank you for the helpful suggestions to improve the manuscript.

Reviewer 2 Report

Comments and Suggestions for Authors

“Triple Antibiotic Combination Exerts Maximally Effective Activity Against M. avium Biofilm and Airway Infection with in vivo Murine Model”

This study seeks to show that an antibiotic cocktail of three antibiotics reduces the carriage of non-tuberculosis mycobacteria both in  vivo and in vitro, and that this cocktail likewise reduces the presence of antibiotic resistant M. avium. While this paper presents reasonable data to support this conclusion, there are multiple overarching issues that should be addressed prior to publication.

Title: M. avium should be completely spelled out in the title.  Additionally, the “maximally effective” superlative seems strong for the ultimate findings and context of this work.

Introduction: The introduction should be thoroughly revised to provide stronger context for the study design and to give more complete background information.  As written, there are some very relevant points made, such as the primary source of NTM infections and the related symptoms of carriage, but these points are not well-connected through the writing to the experiment performed in this work. More background information about the currently applied antibiotics, the success of other cocktail combinations (even if not in NTB), and the known AMR risk NTM presents would all improve the quality of the introduction.  In conjunction with increased context and connection, the introduction could be more transparent about the cocktail/product being tested.  The clinical trial and antibiotic product information is critical information for supporting why this study is being done, but the connection between this study at hand and the companies/products behind it are not well made.  Overall, please improve depth, clarity, and transparency of this introduction through clearer experimental connections and increased background context.  

Results and Methods: The results and methods are generally clear but some missing information means they need to be thoroughly edited for completeness.  CFU is used frequently throughout these sections, but in multiple locations it appears to be incomplete.  CFU is more commonly reported as CFU/item (gram, mL, etc) but such a qualifier is missing throughout. Brand information is missing for multiple materials, or in places where it is provided, it is incomplete.  For instance, GraphPad Prism is referred to by changing nomenclature and missing company location as is more typical for methods sections.  Within these sections please also clarify which isolates were used for what experiments (mice, biofilm, etc) and explain further, possibly in the discussion, why the use of different isolates.  

Figures: The figures with basic X-Y graphs (Like Figure 1) contain hard to read X-axes due to the vertical positioning of the labels.  If possible, please tilt the X-axis labels for easier reading.

Table 3: I find it hard to make comparisons between the treatment groups given the layout of the table, primarily the list format of histopathology observations broken down by treatment group. A matrix set-up would be more amenable to making inter-group comparisons.  Sorting and categorizing by not only treatment group but observation type (congestion, lymphatic findings, bacterial infiltration) would improve this table.

Transparency declaration: I notice there are two listed companies (Certara and RedHill) in the affiliations but only one is directly mentioned in the transparency declaration.  Please double check that this is not an oversight.

Line by Line Comments:

Lines 74-77: The ending of the introduction leaves it unclear why this study is novel or of particular significance and not just a repeat of previous work.  Need to expand on previous work more to prove significance.   

Line 85: resistant should be resistance

Line 87-89: “Bacterial suspensions were generated in Hank’s balanced salt solution (HBSS) and the optical density was used to obtain 3 x 108 CFU/mL.” Please clarify if the OD was taken or if the OD was adjusted.

Line 107: “The final inoculum was prepared to a bacterial density of 105.”  Please clarify if this was done via OD adjustment like mentioned in above methods section or via some other mechanism like plating.  Also add the unit.

Line 113: Add units to 108 bacterial count.  

Lines 113-114: Bacteria were diluted in what?  Please also add the corresponding CFU/mL estimation for the listed OD range.

Line 137: Elaborate on the “four or seven” days.  Why those particular days and why the range?

Line 141: Need to be consistent with the reference to GraphPad (Prism or GraphPad, mixed terminology used throughout).  Need to also list company location: GraphPad (Location)

Line 175-183: The tense shifts from past to present and back in this paragraph.

Table 2: Clarify in the table notes how the percentage values were achieved.

Line 313: Missing a space in M. avium

Author Response

Title: M. avium should be completely spelled out in the title.  Additionally, the “maximally effective” superlative seems strong for the ultimate findings and context of this work.

Answer: We agree with the comments. The title has been modified accordingly.

Introduction: The introduction should be thoroughly revised to provide stronger context for the study design and to give more complete background information.  As written, there are some very relevant points made, such as the primary source of NTM infections and the related symptoms of carriage, but these points are not well-connected through the writing to the experiment performed in this work. More background information about the currently applied antibiotics, the success of other cocktail combinations (even if not in NTB), and the known AMR risk NTM presents would all improve the quality of the introduction.  In conjunction with increased context and connection, the introduction could be more transparent about the cocktail/product being tested.  The clinical trial and antibiotic product information is critical information for supporting why this study is being done, but the connection between this study at hand and the companies/products behind it are not well made.  Overall, please improve depth, clarity, and transparency of this introduction through clearer experimental connections and increased background context.  

Answer: We appreciated the comments. The introduction of the paper has been changes to address the important points raised. It is crucial to transmit the importance of simplified approach for a therapy regimen that last for one or one and a half years, in terms of compliance and potential antibiotic resistance.

Results and Methods: The results and methods are generally clear but some missing information means they need to be thoroughly edited for completeness.  CFU is used frequently throughout these sections, but in multiple locations it appears to be incomplete.  CFU is more commonly reported as CFU/item (gram, mL, etc) but such a qualifier is missing throughout. Brand information is missing for multiple materials, or in places where it is provided, it is incomplete.  For instance, GraphPad Prism is referred to by changing nomenclature and missing company location as is more typical for methods sections.  Within these sections please also clarify which isolates were used for what experiments (mice, biofilm, etc) and explain further, possibly in the discussion, why the use of different isolates.  

Answers: A. The CFU reported has been changed to specify the the denominator. B. The brand information was missing thought out of the methods as pointed out. We ratified the problem. C. The clarification regarding the isolates has been revised.

Figures: The figures with basic X-Y graphs (Like Figure 1) contain hard to read X-axes due to the vertical positioning of the labels.  If possible, please tilt the X-axis labels for easier reading.

Answers: Thank you. While we can appreciate the issue, we believe that the information should be in the axes.

Table 3: I find it hard to make comparisons between the treatment groups given the layout of the table, primarily the list format of histopathology observations broken down by treatment group. A matrix set-up would be more amenable to making inter-group comparisons.  Sorting and categorizing by not only treatment group but observation type (congestion, lymphatic findings, bacterial infiltration) would improve this table.

Answer: Thank you bringing the point out. After examining the possible ways to report the data, we came to the conclusion that way the table is formed, although not perfect, in provides a complete comparison betwwen experimental groups.

Transparency declaration: I notice there are two listed companies (Certara and RedHill) in the affiliations but only one is directly mentioned in the transparency declaration.  Please double check that this is not an oversight.

Answer: The reviewer was correct, and the affiliation of a author with Certara has been included.

Line by Line Comments:

Lines 74-77: The ending of the introduction leaves it unclear why this study is novel or of particular significance and not just a repeat of previous work.  Need to expand on previous work more to prove significance.   

Answer: A statement about the reason for the study has been added to the end of the introduction, as suggested.

Line 85: resistant should be resistance

Asnwers: It has been modified. Thanks.

Line 87-89: “Bacterial suspensions were generated in Hank’s balanced salt solution (HBSS) and the optical density was used to obtain 3 x 108 CFU/mL.” Please clarify if the OD was taken or if the OD was adjusted.

Answer: In all the cases a statement about the OD has been included.

Line 107: “The final inoculum was prepared to a bacterial density of 105.”  Please clarify if this was done via OD adjustment like mentioned in above methods section or via some other mechanism like plating.  Also add the unit.

Answer: The statement about OD measure was added, as well as the unit to the bacterial suspension.

Line 113: Add units to 108 bacterial count.  

Answer: Unit was added.

Lines 113-114: Bacteria were diluted in what?  Please also add the corresponding CFU/mL estimation for the listed OD range.

Answer: The information requested has been added to the text.

Line 137: Elaborate on the “four or seven” days.  Why those particular days and why the range?

Answer: The results are shown for the seven days incubation. The text has been corrected.

Line 141: Need to be consistent with the reference to GraphPad (Prism or GraphPad, mixed terminology used throughout).  Need to also list company location: GraphPad (Location)

Answer: It has been addressed as requested. Thank you.

Line 175-183: The tense shifts from past to present and back in this paragraph.

Answer: Thank you. The issue has been addressed.

Table 2: Clarify in the table notes how the percentage values were achieved.

Answer: The information in table to has been revised to explain how the percentage values were achieved.

Line 313: Missing a space in M. avium

Answer: It has been corrected. Thank you.

Round 2

Reviewer 2 Report

Comments and Suggestions for Authors

The Authors have made clear attempts to address the concerns raised in the first round of review and have improved the quality of the paper overall.  There remain many minor errors throughout (grammatical, spelling, and formatting), but the underlying work is solid and generally clearly presented.  

Comments on the Quality of English Language

Thank you for addressing many of the more minor concerns that were brought up in the previous round of review.  There are still one or two places were CFU lacks a denominator (such as mL, organ, etc). M. avium is also without a space between "m" and "a" in multiple locations. Some of the newly added text also contains grammatical errors, such as in line 172-174: "is one of problems". These errors should be fixed for a more polished presentation of the work prior to publication. 

Author Response

Response to the comments by Reviewer 2

We appreciate the points made by the reviewer 2 about the revised manuscript.

  1. The lack of clarity regarding the CFU denominator:

A: We found 2 locations in which the CFU was not specified regarding the unit. These have been corrected as requested. Thank you.

  1. Space between M and a in referring to M. avium infection.

A: The Reviewer was correct, and we found many examples of that. The issue has been addressed in the revision.

  1. Grammatical awkwardness in some of the new text introduced in the first revision.

A: Thank you for calling our attention about those problems. The sentences have been modified and edited to correct the issues.